# Prenatal Octamethylcyclotetrasiloxane Exposure Impaired Proliferation of Neuronal Progenitor, Leading to Motor, Cognition, Social and Behavioral Functions

**DOI:** 10.3390/ijms222312949

**Published:** 2021-11-30

**Authors:** Dinh Nam Tran, Seon-Mi Park, Eui-Man Jung, Eui-Bae Jeung

**Affiliations:** 1Laboratory of Veterinary Biochemistry and Molecular Biology, Veterinary Medical Center and College of Veterinary Medicine, Chungbuk National University, Cheongju, Chungbuk 28644, Korea; mr.tran90tb@gmail.com (D.N.T.); qkrtjsal0321@naver.com (S.-M.P.); 2Laboratory of Molecular Developmental Biology, Department of Molecular Biology, College of Natural Sciences, Pusan National University, Busandaehang-ro, 63beon-gil 2, Geumjeong-gu, Busan 46241, Korea; jungem@pusan.ac.kr

**Keywords:** cyclic siloxane octamethylcyclotetrasiloxane, endocrine-disrupting chemical, brain development, behavior, CDK6, p27, estrogen receptor

## Abstract

Cyclic siloxane octamethylcyclotetrasiloxane (D4) has raised concerns as an endocrine-disrupting chemical (EDC). D4 is widely used in detergent products, cosmetics, and personal care products. Recently, robust toxicological data for D4 has been reported, but the adverse effects of D4 on brain development are unknown. Here, pregnant mice on gestational day 9.5 were treated daily with D4 to postnatal day 28, and the offspring mice were studied. The prenatal D4-treated mice exhibited cognitive dysfunction, limited memory, and motor learning defect. Moreover, prenatal D4 exposure reduced the proliferation of neuronal progenitors in the offspring mouse brain. Next, the mechanisms through which D4 regulated the cell cycle were investigated. Aberrant gene expression, such as cyclin-dependent kinases CDK6 and cyclin-dependent kinase inhibitor p27, were found in the prenatal D4-treated mice. Furthermore, the estrogen receptors ERa and ERb were increased in the brain of prenatal D4-treated mice. Overall, these findings suggest that D4 exerts estrogen activity that affects the cell cycle progression of neuronal progenitor cells during neurodevelopment, which may be associated with cognitive deficits in offspring.

## 1. Introduction

The effects of environmental pollutants on the brain are raising concern because the brain is sensitive to toxic chemicals and physical stimuli, particularly during development. In normal brain development, the expansion of neural progenitor reservoirs and their differentiation to generate specific neuronal subtypes are important events under the control of several extrinsic and intrinsic factors. Therefore, the regulation of neural progenitor cell (NPC) proliferation and survival is essential to ensure proper brain development. Moreover, neural development is fundamentally linked to the cell cycle [1]. The failure of cell cycle regulation and the survival of NPCs might be the root causes of several neurodevelopmental disorders [2,3]. Moreover, the dysregulation of cell cycle progression lies at the heart of many neurodegenerative disorders, such as Alzheimer’s Disease and Parkinson’s Disease [4]. The four sequential stages (G1, S, G2, and M) of cell cycle progression are positively regulated by complexes of cyclin-dependent kinases (CDKs) and associated cyclins, which are negatively regulated by cyclin-dependent kinase inhibitors (CKIs). Mammalian CKIs are divided into two families based on evolutionary origins, structure, and CDK specificities: the INK4 family and the Cip/Kip family [5]. The INK4 gene family, including p15, p16, p18, and p19, bind to CDK4 and CDK6 and inhibit their kinase activities by interfering with their association with D-type cyclins. In contrast, the Cip/Kip family members p21, p27, and p57 bind to both cyclin and CDK subunits and can modulate the activities of cyclin D-, E-, A-, and B-CDK complexes [6]. Among the CKIs, p27 is expressed in the developing brain and plays an important role in neurogenesis by promoting the cell cycle arrest of neural progenitors [7,8]. Moreover, p27 is involved in cell cycle withdrawal and neuronal migration [9]. The protein level of p27 is downregulated rapidly as the cells enter the cell cycle [10].

Humans are exposed to organosiloxane compounds because of their presence in many products used daily, such as deodorants, skin and hair care products, drugs, and water repellants. The cyclic volatile methyl siloxanes (cVMS) are a class of silicone compounds, in which octamethylcyclotetrasiloxane (D4) is a low molecular weight synthetically derived silicone fluid that is clear and odorless. Thus, D4 is used primarily as an intermediate in the manufacture of high molecular weight silicone polymers or vehicles or ingredients in consumer and precision cleaning products. On the other hand, D4 is a silicon-based material with octanol-water partitioning coefficients of 6.98 [11] and water solubility of 17 μg/L [12]. Hence, D4 can be released into the aquatic environment. Therefore, there is increasing concern about the fate and effects of D4 [13]. D4 is an endocrine-disrupting chemical (EDC) with estrogenic potential [14]. Chronic exposure to D4 increased the liver, kidney, testes, and uterine weight with correlating morphological changes in rats [15]. Exposure to D4 resulted in a dose-dependent increase in uterine weight and epithelial cell height [16]. On the other hand, the effects of D4 on brain development are unclear. Early exposure to environmental pollutants, such as EDCs, affects cell differentiation, proliferation, and maturation, altering normal development. Moreover, early exposure to EDCs was recently reported to result in neurodevelopmental disorders, leading to abnormal behavior [17,18]. These studies also showed that EDCs affected the cell cycle and survival of dentate gyrus (DG) neural progenitors in embryonic and adult neurogenesis. However, the possible molecular mechanisms through which EDCs impair brain development are unknown.

In this study, prenatal exposure to D4 induced the strong upregulation of p27 and downregulation of CDK6 in neuronal progenitors by regulating the estrogen receptors, resulting in impaired cell proliferation. These caused strong cell cycle arrest, accompanied by apoptotic and cytotoxic activity. Furthermore, prenatal D4 treated mice exhibited abnormal behaviors.

## 2. Results

### 2.1. Cytotoxic Effects of D4 on Sox1-GFP Mouse Embryonic Stem Cells and Primary Cortical Neuronal Culture

The neurotoxicity effects of D4 were investigated by determining two endpoint values, including IC50 for cell viability/growth and ID50 for cell differentiation. Among the various mouse embryonic stem cells (MESC), the Sox1-GFP knock-in (46C), in which enhanced green fluorescence protein (GFP) is inserted into the Sox1 locus, was used. Sox1 is the earliest known marker for committed neural precursors in the proliferating neuroepithelium, and Sox1-GFP expression is likely to identify an earlier subset of neural precursors. Therefore, the Sox1-GFP reporter system is highly useful for the identification, isolation, and characterization of neural stem and progenitor cells [19]. The cell viability of the Sox1-GFP mouse embryonic stem cell lines was first measured using CCK8 assays after treatment with various concentrations of D4 (10^−12^–10^−4^ M) for 48 h. The viability of Sox1-GFP cells decreased significantly upon a D4 treatment with IC50 values of 3.56 × 10^−7^ M (Figure 1A). In addition, the ID50 value of D4 was determined to be the 50% reduction in intensity of integral GFP in the neurosphere area relative to control. The ID50 value was 2.23 × 10^−6^ M (Figure 1B).

Based on these data, the effects of D4 on the proliferation and survival of neuronal progenitor cells using a primary cortical neuronal cell culture were examined. Primary cortical neurons were exposed to D4 at concentrations of 10^−8^–10^−5^ M from the day in vitro (DIV) 1 to 5. The percentage of BrdU^+^ cell was markedly lower after a 12 h treatment with D4 at a dose of 10^−6^ M and 10^−5^ M than in the vehicle, D4 10^−8^ M, and D4 10^−7^ M groups (Figure 1C,E). The D4 10^−8^ M and D4 10^−7^ M groups exhibited no significant decrease in the percentage of BrdU^+^ cells compared to the vehicle group (Figure 1C,E). In addition, the percentages of cleaved caspase-3^+^ cells were slightly higher in the D4 10^−7^ M, D4 10^−6^ M, and D4 10^−5^ M groups than in the vehicle group (Figure 1D,F). The percentage of cleaved caspase-3^+^ cells in the D4 10^−8^ M group was similar to the vehicle group (Figure 1D,F). The morphology of the primary cortical neurons was observed by microtubule-associated protein-2 (MAP2) staining and Tau1 to determine the average lengths and numbers of axons and dendrites on DIV 5 (Figure 1G). The numbers of primary and secondary axons and dendrites were similar in the D4-treated and VE groups (Figure 1H–L). Furthermore, the D4-treated groups showed no difference in the average axon and dendrite lengths to those of the VE group (Figure 1J,M). These results suggested that D4 might inhibit proliferation and promote the apoptosis of neuronal progenitor cells but not impair the growth of neurons during the early stages of differentiation.

### 2.2. Prenatal D4-Treatment Reduced the Survival of Pups

To evaluate the effects of prenatal D4 on neurodevelopment, pregnant dams were treated with D4 at doses of 10, 25, or 50 mg/kg from E 9.5 to PND 28. The bodyweight of the D4-treated dams showed no significant difference during the pregnant period (Figure 2A), and there was no significant difference in the bodyweights of newborn pups on PND1 between the vehicle and D4-treated groups (Figure 2B). The number of pups was also similar in the vehicle and D4-treated groups (Figure 2C). On the other hand, the D4 50 mg/kg group showed a reduced survival of pups compared to the vehicle group (Figure 2D). The survival of pups was similar in the D4 10 mg/kg and D4 25 mg/kg groups compared to the vehicle group (Figure 2D). No change in the bodyweight of the offspring mice was observed on PND 28 and 56 (Figure 2E). The brain weight in the D4 50 mg/kg group was lower than the vehicle group (Figure 2F). D4 10 mg/kg and D4 25 mg/kg showed a similar brain weight to the vehicle group (Figure 2F). These results suggest that prenatal D4 treatment induced abnormal brain development.

### 2.3. Prenatal D4 Treatment Induced Behavioral Deficits in Adulthood

Initially, the postpartum female mice exposed to D4 showed normal maternal behavior, including pup retrieval, maternal aggression, and milk spots in the stomach of offspring mice (data not shown). The behavioral performance of the six-week-old vehicle and D4-treated mice was tested to examine the effects of prenatal D4 exposure on behavior in adulthood. The Morris water maze test was first performed. During the training phase (four trials per day for four successive days), although the vehicle and D4-treated groups showed improved latency to the platform, the D4 25 mg/kg and 50 mg/kg groups showed a significant delay in finding the platform (Figure 3A). The D4 25 mg/kg group displayed a higher escape latency time than the vehicle group two days after training (Figure 3A). Moreover, the differences in escape latency time between the D4 50 mg/kg and vehicle groups continued consistently after four days of training (Figure 3A), indicating that the capacity for spatial learning in the D4 50 mg/kg group was limited, rather than delayed. In the subsequent probe test phase, the D4 50 mg/kg group crossed the platform location fewer times than in the vehicle group (Figure 3C). There was no difference in the platform crossing time between the vehicle and D4 10 mg/kg, D4 25 mg/kg groups (Figure 3C). On the other hand, the D4 50 mg/kg group spent less time in the area where the platform was initially located than the control group (Figure 3D). Visual representation of the results showed that the D4 50 mg/kg group had lower proximities to the old platform quadrant than the vehicle group (Figure 3B). Therefore, the vehicle and D4-treated mice showed no differences in swimming distances and swimming speeds (Figure 3E,F). In the novel object recognition test to investigate the effects on recognition memory, the vehicle and D4 10 mg/kg groups spent more time approaching and in proximity to the novel object (Figure 3G). The D4 25 mg/kg and D4 50 mg/kg groups, however, displayed no preference for exploring either familiar or novel objects (Figure 3G). Furthermore, in the rotarod test for motor learning, the vehicle and D4-treated mice showed an increase in the latency to fall during the three-day training phase (Figure 3H). In contrast, the D4-treated groups did not show the same increase in the latency to fall than the controls during the training phase (Figure 3H). Throughout the testing phase (days 4 to 20), both the vehicle and D4-treated mice showed a consistent latency to fall. The D4 25 mg/kg and D4 50 mg/kg groups showed a lower latency to fall than vehicle and D4 10 mg/kg groups during the training phase (Figure 3H), suggesting deficits in motor learning ability. Grip strength tests were performed for a more specific measure of the muscle strength. The grip strength of the D4-treated mice was similar to the vehicle mice (Figure 3I). These results show that prenatal exposure to D4 has impaired cognitive functioning in spatial and nonspatial learning and memory.

The sociability and social novelty of D4-treated mice were next evaluated using the three-chamber test. In the sociability test, all vehicle and D4-treated mice spent more time in the chamber containing an unfamiliar mouse than in the empty chamber (Figure 3J). The preference index for social interactions was similar in the VE and D4-treated groups (Figure 3K). In the social novelty test, vehicle and D4 10 mg/kg groups showed a preference for the second unfamiliar mouse. In contrast, the D4 25 mg/kg and D4 50 mg/kg groups showed a similar preferred time in the chamber containing the first, more familiar mouse (Figure 3L). Moreover, the preference index of the novel stimulus was markedly lower in the D4 50mg/kg group than in the VE group (Figure 3M). The D4 10 mg/kg and 25 mg/kg groups also showed a lower preference index for the novel stimulus than the VE group, but the difference was not significant (Figure 3M). These results show that prenatal exposure to D4 impaired the discrimination of social novelty. A forced swim test was also performed to assess depression-related behavior. The D4 25 mg/kg and D4 50 mg/kg groups displayed markedly increased immobilization behavior than the VE group (Figure 3N). Moreover, the D4-treated groups buried more marbles than the vehicle groups in the marble-burying test (Figure 3O). Furthermore, using the open field test, the D4-treated mice showed no elevated anxiety-like behavior (Appendix A). On the other hand, the D4 25 mg/kg and D4 50 mg/kg groups showed longer moving distances than the VE group (Appendix A). The D4-treated groups displayed no change in the velocity compared to the vehicle group (Appendix A). The incidence of repetitive, stereotyped behaviors was next examined by looking at self-grooming bouts and performing the nesting test. The D4-treated mice showed no change in the grooming time and nesting score (Appendix A). Together, these data suggest that prenatal exposure to D4 also led to intellectual functioning impairments and specific social deficit elevated depression-related behaviors.

### 2.4. Prenatal D4 Impairs Neural Progenitor Cell Premature Differentiation and Maintenance In Vivo

The effects of D4 treatments on neurodevelopment were examined to reveal the cause of these behavioral deficits. First, the effects of perinatal exposure to D4 on the proliferation of precursor neuronal cells during embryogenesis were evaluated. BrdU was injected into the D4-treated and vehicle embryos at E17.5 by injecting pregnant females and harvesting embryos 2 h later. The number of S-phase cells in the dentate gyrus (DG) area was determined, enabling the detection of cells undergoing DNA replication at that time (Figure 4A). The numbers of BrdU^+^ cells were slightly lower in the E18.5 of D4 50 mg/kg group than in the vehicle group (Figure 4B). There was no significant difference in the number of BrdU^+^ cells between the vehicle, D4 10 mg/kg, and D4 25 mg/kg groups (Figure 4B). In addition, the number of cells expressing the mitotic marker, Ki67^+^ at DG, was significantly lower in the D4 50 mg/kg group than the vehicle group (Figure 4A,C). On the other hand, the D4 10 mg/kg and D4 25 mg/kg groups showed a similar number of Ki67^+^ cells to the vehicle group (Figure 4C).

In the offspring mice, BrdU was injected in the D4-treated and vehicle adults, and the brain was collected 2 h later. The number of BrdU^+^ cells in the D4 10 mg/kg group was similar to that of the vehicle group (Figure 4D,E). On the other hand, the D4 25 mg/kg and D4 50 mg/kg groups showed a slightly lower number of BrdU^+^ cells than in the vehicle group (Figure 4E). Furthermore, D4 50 mg/kg group had a significantly lower number of Ki67^+^ cells than the vehicle group (Figure 4D,F). No difference in the number of Ki67^+^ cells was found between the vehicle group, D4 10 mg/kg, and D4 25 mg/kg groups (Figure 4F). These data suggested that a perinatal D4 treatment impairs neurogenesis.

The proportion of cell cycle exits (BrdU^+^Ki67^−^/BrdU^+^) was increased significantly in the D4 25 mg/kg and D4 50 mg/kg groups than the vehicle groups in both E17.5 and adult DG (Figure 4G–L). Moreover, a significantly lower proportion of cells was observed to re-enter the cycle in the D4 25 mg/kg and D4 50 mg/kg groups than the vehicle group (BrdU^+^Ki67^+^/BrdU^+^) (Figure 4G–L). The proportion of cell cycle exits and cell cycle re-entries was similar in the D4 10 mg/kg, vehicle, D4 25 mg/kg, and D4 50 mg/kg groups in E17.5 and adult DG (Figure 4G–L). These suggest that prenatal exposure to D4 affects the cell cycle progression of DG neural progenitors in embryonic and adult neurogenesis.

### 2.5. Prenatal D4 Impaired Neurogenesis but Not the Neural Migration

In the cerebral cortex, neurons were generated in the cortical ventricular zone (VZ) and subventricular zone (SVZ), which migrate radially to their appropriate position, resulting in the formation of cortical layers in an inside-out manner [20]. Moreover, a defect in neuron migration led to reduced neuronal cells [21]. This study verified whether the D4 treatment impaired neural migration. The hypothesis was that the ectopic distribution of neurons in the cortex would be disrupted if D4 impaired neural migration. A migration assay was performed using BrdU, through which newborn cells can be labeled because it is incorporated into the newly synthesized DNA [22]. Pregnant mice were single injected on E12.5 with BrdU, and the BrdU^+^ cells in the brains of offspring were counted in adulthood (8–10 weeks old) (Figure 5A). The BrdU^+^ cells were distributed normally in the cerebral cortex with an inside-out pattern, and there were no apparent differences between the VE and D4 mice (Figure 5B,C). On the other hand, the number of BrdU^+^ cells decreased significantly in the D4-treated groups than in the VE group (Figure 5D). The D4 10 mg/kg, D4 25 mg/kg and D4 50 mg/kg groups had a similar number of BrdU^+^ cells (Figure 5D). These results suggest that prenatal D4 impaired neurogenesis not neural migration.

### 2.6. Prenatal D4 Impairs Cell Cycle Regulatory Protein in Neural Progenitor Cells

Cyclins and CDKs tightly regulate cell cycle progression in eukaryotes. CKIs and the INK4 inhibit the cyclin/CDKs complexes. In this study, the levels of cyclin D1 and D3 in the DG were similar in the D4 and vehicle mice at both E17.5 and adult stages (Figure 6A,B). On the other hand, the protein levels of CDK6 were lower in the D4 50 mg/kg group than in the vehicle group at the E17.5 and adult stages (Figure 6A–C). There was a significant decrease in the level of the CDK6 protein in the D4 25 mg/kg group at E17.5, but not at the adult stage compared to the vehicle group (Figure 6A–C). Moreover, the level of the CDK6 protein in the D4 10 mg/kg group was markedly lower than that in the D4 25 mg/kg and D4 50 mg/kg groups at the embryo DG (Figure 6A,B).

The levels of CKIs were analyzed. The D4 treatment induced an increase in the p27 protein levels in the DG of embryos compared to the vehicle group (Figure 6A,B). A slightly higher p27 protein level was found in the adult DG of the D4-treated groups than in the vehicle group (Figure 6A,C). The p27 protein levels were similar in the D4 10 mg/kg, D4 25 mg/kg, and D4 50 mg/kg groups at both E17.5 and adult stages (Figure 6C). These results suggest that decreases in CDK6 and increases in p27 play important roles in the D4-induced arrest of neuronal progenitor cell growth.

### 2.7. Prenatal D4 Increases ERa and Decreases ERb Expression of the Hippocampus in Mice

D4 binds to ERa and exerts estrogen-like effects [23]; thus, ERa may be involved in the D4 effects. Prenatal exposure to D4 resulted in an increase in uterine weight compared to the vehicle group (Appendix A). Furthermore, the protein levels of ERa in the hippocampus at E17.5 were significantly higher in the D4-treated groups than the vehicle group (Figure 6A,B). In the adult hippocampus, the protein levels of ERa were significantly higher in the D4 25 mg/kg and D4 50 mg/kg groups than the vehicle groups (Figure 6A,C). The D4 10 mg/kg group showed a similar ERa protein level to the vehicle and D4 25 mg/kg groups (Figure 6A,C). On the other hand, the ERa protein level in the D4 50 mg/kg group was higher than in the D4 10 mg/kg group at the adult stage (Figure 6A,C). ERa is the main ER isoform in regulating neuronal development associated with affective behaviors and cognition [24]. Although D4 binds to ERa but not ERb, the expression of both ERa and ERb proteins were also increased by perinatal exposure to D4 (Figure 6A–C). A significant difference was found in the D4 25 mg/kg and D4 50 mg/kg treatment groups than the vehicle group in DG at both the embryonic and adult stage (Figure 6A–C). The ERb protein levels in the D4 10 mg/kg groups were similar to the vehicle, D4 25 mg/kg, and D4 50 mg/kg treatment groups (Figure 6A–C). These findings highlight the potential roles of ERa and ERb in D4-induced pathological effects.

### 2.8. Principal Component Analysis (PCA)

PCA analysis was performed to interpret the relationships between brain weight, behavior analysis data, neurogenesis analysis parameters, and estrogenic capacity in mice subjected to different D4 treatments (Figure 7). The first four principal components (PCs) were associated with eigenvalues > 1 and explained 94.5% of the cumulative variance, with PC1 accounting for 35.8%, PC2 for 13.2%, PC3 for 10.3%, and PC4 for 8.0%. PC1 was positively correlated with ERa, ERb, p27, cell cycle exit, brain weight, and behavior tests, including Morris, social, forced, and marble (Figure 7A). PC1 was also negatively correlated with cell cycle re-entry, CDK6, and BrdU (Figure 7A). Moreover, PC2 was negatively correlated with Ki67, brain weight, rotarod test, and social test (Figure 7A). The loading matrix also indicates the correlations between the examined quanti-qualitative traits. The longer vector length of the parameters could represent PC 1 or PC 2 better (Figure 7A). Moreover, the narrow angle between the parameters depicted a significant positive correlation; the obtuse angle reflected a significant negative correlation, and the right angle described no relationship. In the present study, the variations in ERb showed stronger positive correlations with cycle exit and p27 and negative correlations with cycle re-entry, CDK6, and BrdU than the ERa variations (Figure 7A), suggesting that ERb mainly involves the effects of D4 on brain development. Clustering of the data points was seen at each D4 dose, and the clusters were aligned along the horizontal axis in the PCA scatter plot (Figure 7B). This indicated that PC 1 strongly correlates with the dose of the D4 treatments. Moreover, PC1 was negatively correlated with the vehicle and D4 10 mg/kg groups but positively correlated with the D4 25 mg/kg and D4 50 mg/kg groups (Figure 7B). These results suggest that the effects of D4 on brain development are dose-dependent.

## 3. Discussion

This study examined the mechanism through which D4 impaired brain development and behavior. D4 regulated the cell cycle machinery by affecting both the growth-promoting and growth-inhibiting aspects of cell cycle progression. The key to the ability of D4 to arrest NPC growth was the regulation of CDK6 and p27 expression. A prenatal treatment with D4 decreased CDK6 and increased p27 expression via estrogen activity.

In a previous study, mice with prenatal exposure to D4 showed no significant fetal malformations or developmental variations [25]. In the present study, prenatal exposure to D4 resulted in a normal number and bodyweight of newborn pups. On the other hand, after birth, the survival of the pups and the adult brain weight were reduced by prenatal D4 exposure. These findings suggest that the effects of D4 increased gradually with growth. In adulthood, mice with prenatal exposure to D4 exhibited abnormal behavior, such as cognitive dysfunction, intellectual functioning impairments, and specific social deficit elevated depression-related behavior. Moreover, the decrease in the weight of the brain in adults strongly indicated that the prenatal D4 treatment impaired neurodevelopment.

Brain development is a remarkably well-organized process involving many events, including neurogenesis, neuronal migration, synaptogenesis, gliogenesis, and neuronal wiring. These events are crucial in terms of their sequence and timing. During the neurogenesis process, new neurons are generated from neural stem cells or progenitor cells through asymmetric and symmetric divisions. In the early stages, neural progenitor cells are divided by the symmetrical model of cell division, producing two identical neural progenitor cells [26]. The neuronal progenitor cells in the VZ and radial glial cells (RGC) begin to shift the cell division mode from symmetrical to asymmetrical, which produces two daughter neurons, including one neuronal progenitor cell and one postmitotic neuron cell [27]. During asymmetric division, the new progenitor cell remains in the VZ as an RGC and continues to divide. Simultaneously, the other becomes either a postmitotic neuron or an intermediate neuronal progenitor cell [28]. These neurons migrate radially from the VZ to their target region in the CNS, resulting in the formation of a layered structure with an inside-out pattern. The prenatal D4 treatment reduced the number of BrdU^+^ cells in the early developmental and adult stages. These findings suggest that prenatal D4 exposure could affect the proliferation of the neuronal progenitors in the fetal and adult brain, resulting in an impairment of neurodevelopment. In this study, however, the prenatal D4 treatment did not affect migration. These effects of D4 on the proliferation of neural progenitors are consistent with a previous study with other EDCs, such as Bisphenol A and Triclosan [17,18]. However, the possible molecular mechanisms through which EDCs impair brain development are unknown.

The proliferation and growth arrest of neuronal progenitors is regulated by a balance of extrinsic and intrinsic signals that direct the entry, progression, and exit from the cell cycle. Moreover, the number of neurons derived from the progenitor cells is determined by two cell cycle parameters: the rate of cell cycle progression and the balance between cell cycle re-entry and exit [20]. A dysregulation of the cell cycle in proliferating cells impacts not only the number of neurons, cytoarchitecture, and brain development but also adult behavior [29]. Furthermore, control of the proliferation of neuronal precursor cells plays a crucial role in determining the number of neurons during brain development. The major regulatory events leading to proliferation occur in the G1 phase of the cell cycle. In general, quiescent cells are stimulated by growth factors to synthesize D-type cyclins (D1, D2, and D3) and initiate cell division. Within the early embryo, cyclins D1 and D2 show specific expression patterns in the developing brain [30]. Moreover, cyclin D1 is associated with the G1-S phase transition [31]. Cyclin D1 is found in neural progenitor cells during mouse brain development [32]. In addition, the loss of cyclin D1 reduces the proliferation of granule neuron precursors, leading to an impairment in the growth of the cerebellum [33]. On the other hand, the present study showed that prenatal exposure to D4 did not alter cyclin D1 and D3 expression. Moreover, cyclin D1 interacts with CDK4/6, in which only CDK6 but not CDK4 was found to be critical for neural progenitor proliferation in the subgranular zone (SGZ) of the hippocampal dentate gyrus and SVZ of the lateral ventricle [34]. The CDK6 activity was observed at the mid G1 phase of the cell cycle and is responsible for G1 progression and G1 to S phase transition. The loss of CDK6 in precursor cells prematurely exiting the cell cycle and a lengthened G1 phase results in a decrease in the production of newborn neurons [35]. In contrast to CDK4/6, p27 is a necessary component of cyclin D-dependent kinase activity; high levels of p27 inhibit cyclin D- and E-dependent kinase activity. Thus, p27 has a negative effect on the G1 phase of the cell cycle. During embryogenesis, p27 has been implicated in promoting the cell cycle arrest of neural progenitors [7]. Moreover, p27 plays essential roles in regulating the division of transit-amplifying progenitors in the adult subventricular zone [36]. In this study, the prenatal D4 treatment downregulated CDK6 expression but upregulated p27 expression. Interestingly, D4 altered only CDK6 expression but not CDK4 (data not shown). Moreover, p27 binds to the cyclin D/CDK6 complexes in cycling cells, suggesting that the downregulation of CDK6 by D4 may help to increase the p27 levels. The D4 treatment-induced decrease in CDK6 may release p27 from the cyclin D/CDK6 complexes. Moreover, PCA analysis showed that expression of CDK6 and p27 strongly contributed to the dysregulation of the cell cycle in the NPCs of D4-treated mice. These results suggest that the decrease in CDK6 and increase in p27 play important roles in the D4-induced arrest of NPCs proliferation during brain development.

D4 is an EDC that can induce estrogen-like activity by binding to the estrogen receptor (ER). Therefore, it may interfere with the actions of the endogenous steroid hormones or induce hormone-mediated responses. Moreover, changes in the relative expression levels of the ER receptors may result in differences in gene expression mediated by EDC. Estrogen is an important hormone that is involved in multiple effects on the neuronal physiology and survival. The estrogenic actions are mediated mainly through two distinct estrogen receptor (ER) subtypes, ERa and ERb. Both ERs are expressed in neurons and glial cells in different brain areas during development. In the brain, ERa is the predominant ER in the hypothalamus, whereas ERb is the main ER expressed in the cerebral cortex, hippocampus, and other brain areas [37]. Moreover, mice lacking ERβ showed impaired corticogenesis [38], increased vulnerability to neurodegeneration [39], increased anxiety-like behavior [40], and behavioral deficits related to impaired spatial learning [41]. ERb can be measured at E10.5, and it precedes ERa expression, which is observed only after E16.5 and the first availability of embryonic estrogen at E18.5 in mice [42]. In NPCs, the ERa mRNA levels were approximately 20-fold lower than the ERb levels [43]. Moreover, the level of ERb increased during the development of NSC, whereas that of ERa decreased [44]. In the present study, prenatal exposure to D4 increased ERa and ERb expression in the DG area of mice. Furthermore, the proliferative activity capacity of adult and embryonic NPCs was significantly lower in the D4-treated mice. Recently, ERb was reported to inhibit proliferation by repressing c-myc, cyclin D, and cyclin A gene transcription and increases the expression of p21 and p27, which leads to cell cycle arrest [45]. These results suggest that ERb is the main ER isoform in regulating neuronal development associated with the affective behaviors and cognition. Furthermore, the expression of ERb showed stronger correlations with the cell cycle, CDK6, and p27 than ERa expression in the D4-treated mice. These findings show that the effects of D4 on the NPC occur mainly through the ERb receptor.

In conclusion, D4 increases the expression of estrogen receptors, inhibits the cell cycle progression of neuronal progenitor cells by regulating the expression of cyclin-dependent kinases CDK6 and cyclin-dependent kinase inhibitor p27. Furthermore, D4 has weak estrogenic activity, and that these effects are mediated through estrogen receptors [23,46]. Therefore, the present findings showed the possible molecular mechanisms through which EDCs impair the brain development of mice. These should encourage thorough consideration of the adverse effects of EDC exposure in humans.

## 4. Materials and Methods

### 4.1. Animals

Specific pathogen-free C57BL/6J male and female mice (8-weeks-old, 25–30 g) were obtained from Samtako (Samtako Bio Korea, Osan, Gyeonggi, Korea), and bred under controlled environment conditions as in previous [47]. After 1 week of the acclimatization period, female mice were mated with male mice overnight at a proportion of 2:1. The day of vaginal plug detection was set as embryonic day (E) 0.5 and the day of birth as postnatal day 0 (PND0). All experimental protocols were approved by the Institutional Animal Care and Use Committee of Chungbuk National University, and all experiments were carried out in accordance with the relevant guidelines and regulations (project identification code: CBNUA-1371-20-02).

The pregnant mice were randomly divided into four groups (*n* = 5 mouse/group, each mouse was maintained in each cage) and daily received a subcutaneous injection of corn oil (vehicle group), or D4 (10 mg/kg/day), or D4 (25 mg/kg/day), D4 (50 mg/kg/day), (D4, Sigma-Aldrich Corp, St. Louis, MO, USA) dissolved in corn oil (Sigma-Aldrich) from E9.5 to postnatal day (PND) 28. After weaning on PND28, offspring were separated by gender, and then randomly assigned to groups for all experiments to be performed at 6–10 weeks of age.

### 4.2. Cell Culture

The Sox1-GFP mESCs (kindly donated by Professor Eekhoon Jho, Cellular Signaling Transduction laboratory, University of Seoul) were cultured in DMEM (Gibco-BRL, Thermo Fisher Scientific, Gaithersburg, MD, USA) with 15% fetal bovine serum (FBS; Biowest, Rue de la Caille, Nuaille, France), L-glutamine (Gibco), thioglycerol (Sigma-Aldrich), 100 units per mL penicillin and 100 µg per mL streptomycin (Biowest), MEM Non-Essential Amino Acids (NEAA; Gibco), and leukemia inhibitory factor (LIF; Sigma-Aldrich) in 96-well plates (Nunc A/S, Roskilde, Sjælland, Denmark) coated with 0.2% gelatin. For neuron differentiation, cells were cultured in DMEM/F12 (Gibco) with bovine serum albumin (BSA) fraction V (7.5%) (Gibco), L-glutamine, 2-mercaptoethanol (Gibco), penicillin/streptomycin, N2 supplement (Gibco), B27 supplement (Gibco), and bovine serum albumin (BSA) fraction V (7.5%) (Gibco). All cells were cultured at 37 °C in air containing 5% CO_2_. Once 75% confluency was achieved, they were treated with various doses of D4 (Sigma-Aldrich) for 48h. D4 (31.03 µL) was reconstituted in 68.97 µL DMSO to 1 M. The final concentration of DMSO in the vehicle and experimental groups was maintained at less than 0.1% in all treatments.

Cell viability assay. Cells were seeded into 96-well culture plates at 7000 cells/well for 48 h. Cell ability was evaluated using Cell Counting Kit-8 (Dojindo Molecular Technologies, Kumamoto, Japan) according to manufacturer’s protocol. After a specified period of drug treatment, cells were rinsed with DPBS and incubated with CCK8 solution for 30 min at 37 °C in air containing 5% CO_2_. The optical density (OD) of each well was measured with a microplate reader Epoch Microplate Spectrophotometer (BioTek Instruments, Winooski, VT, USA) at a wavelength of 450 nm. Normalization was carried out for cells treated with DMSO as the vehicle, which were defined as 100%. The 50% of inhibitory concentration (IC_50_) values were calculated according to the concentration–response curve using GraphPad Prism 5 (GraphPad Prism Software, San Diego, CA, USA).

Neuronal differentiation ability. Cells were cultured with differentiation medium at 100 cells/well, with/without the final concentration of the test chemicals in 96-well round-bottom plates (Corning Incorporated, Corning, NY, USA). After 4 days of culture, inhibition of neuronal differentiation (ID_50_) was determined by 50% reduction in the intensity fluorescence of GFP relative to the vehicle. The ID50 values were calculated according to the concentration-relative curve using GraphPad Prism 5.

Primary cortical neuron culture. Primary cortical neurons were prepared from E15.5 mouse brain as previously described [48]. Briefly, dissected embryonic cortices were trypsinized (Celgene Corporation, Summit, NJ, USA) at 37 °C for 10 min, triturated, and seeded in 24-well plates pre-coated with poly-d-lysine and laminin (both Sigma-Aldrich). In all, 1 × 10^5^ cortical neurons per well were cultured in Neurobasal medium/DMEM (1:1) with B27 supplement, penicillin, streptomycin, and glutamine (Gibco) at 37 °C in a humidified incubator with 5% CO_2_/95% air. The day of plating was considered a day in vitro 0 (DIV 0). On DIV 1, cells received D4 at a range of concentrations 10^−8^ M, 10^−7^ M, 10^−6^ M, and 10^−5^ M. Cortical neurons were harvested for quantification of the axon and dendrite morphology on DIV 4. For proliferative experiments, cortical neurons were incubated with 5-Bromo-2′-deoxyuridine (BrdU) (10 mM/mL, Sigma-Aldrich, St. Louis, MO, USA) after 12 h administering of D4. Cortical neurons were harvest after 2 h BrdU-treatment.

### 4.3. BrdU Injection

Pregnant mice or adult offspring mice were injected intraperitoneally with 50 mg/kg or 100 mg/kg 5-Bromo-20-deoxyuridine (BrdU; Sigma-Aldrich, St. Louis, MO, USA) in saline, respectively.

Migration assay. Pregnant mice were received a single pulse of BrdU injection to label neural progenitors cycling through the S-phase in E13 embryos during D4 exposure. Brains were collected on postnatal week 8–10 (adult) [22].

For proliferation assay. Pregnant mice at E17.5 or adult offspring mice received a single pulse of BrdU 100 mg/kg injection. After 2 h, mice were perfused with 4% paraformaldehyde in PBS (pH 7.4) and brain tissue collected.

### 4.4. Immunofluorescence

Mice were anesthetized with 2.5% Avertin (200 mg/kg body weight; 2,2,2-tribromoethanol: T48402, Sigma-Aldrich; Ter-amyl alcohol: 240486, Sigma-Aldrich) and transcardially perfused with cold PBS followed by 4% paraformaldehyde in phosphate buffer. Brains were removed and briefly fixed in phosphate-buffered 4% paraformaldehyde (PFA) for at 4 °C overnight. Finally, brain transfer to 1× PBS and were sectioned coronally at 80 µm with a microtome (SM2010R, Leica Microsystems Ltd., Leider Lane, Buffalo Grove, IL, USA). Brain sections were stored in 1× PBS and kept at 4 °C until staining.

Cells cultured on coverslips or brain sections were fixed with 4% paraformaldehyde for 15 min at room temperature and permeabilized for 10 min with 0.1% or 0.5% Triton X-100 (Sigma-Aldrich), respectively. Cells or brain sections were blocked with PBS + 5% goat serum (Vector laboratories, Burlingame, CA, USA) + 0.25% Triton X-100) for 1 h at room temperature, and then cells were incubated with primary antibody (MAP2, Abcam, Cambridge, UK, cat. no. ab32454, 1:500; and Tau1, Abcam, cat. no. ab75714, 1:500; Ki67, Cell Signaling Technology, Danvers, MA, USA, cat. no. D385, 1:500; BrdU, Bioscience, Durham, NC, USA, cat. no. 555627, 1:1000) at 4 °C overnight. The cell or brain sections were washed three times with PBS for 10 min each time, and then incubated with secondary antibody solution (Alexa Fluor488 goat anti-rabbit IgG, cat. no. A11034, 1:1000, Alexa Fluor594 goat anti-rabbit, cat. no. A11012, 1:1000 and Alexa Fluor488 goat anti-chicken, cat. no. A11039, 1:1000, Invitrogen, Thermo Fisher Scientific, Carlsbad, CA, USA) that contained 100 ng/mL 4′,6-diamidino-2-phenylindole (DAPI) (Sigma-Aldrich) for 1 h at room temperature and protected from light. Finally, cells or brain sections were mounted in Fluoro-Gel (Emsdiasum, Electron Microscopy Sciences, Hatfield, PA, USA) and observed using a BioTek Lionheart FX microscope (BioTek Instruments).

### 4.5. Western-Blot Analysis

Total protein content was extracted using Pro-prep solution (iNtRON Biotechnology Inc., Seoul, Korea) according to the manufacturer’s protocol. Equal amounts of protein (50–100 µg) were resolved by using 12% sodium dodecyl sulfate-polyacrylamide gel electrophoresis and transferred to polyvinylidene fluoride membrane (Merck Millipore, Taunton, MA, USA) as previously described [49]. Thereafter, membranes were blocked with 5% milk at RT for 1 h. Then, membranes were incubated overnight with primary antibodies (Cyclin D1, Cyclin D3, CDK6, p27, Cell Signaling Technology, cat. no. 9932T, 1:500; ERa, Santa Cruz Biotechnology Inc, Santa Cruz, CA, USA, cat. no. sc-8002, 1:1000, ERb, Santacruz, cat. no. sc-373853, 1:1000) and secondary antibodies (anti-rabbit, Cell Signaling Technology, 1:3000; anti-mouse, Cell Signaling Technology, 1:3000). Membranes were enhanced using chemiluminescence reagent (EMD Millipore Corporation, Burlington, MA, USA). Band intensities were detected with the Chemi Doc equipment, GenGnome5 (Syngene, Cambridge, UK) and quantified by using Image J software (National Institutes of Health, Bethesda, MD, USA). Target protein band intensities were normalized to the band intensities of the housekeeping gene β-actin.

### 4.6. Behavioral Analysis

At 6 to 10 weeks of age, offspring mice were randomly selected to test for mood and cognitive function using a variety of behavioral analyses as described in previous studies [48,50]. Please see Supplemental Methods in Appendix A for a full description of behavioral assays.

### 4.7. Statistics and Principal Component Analysis

All data were subjected to analysis of applying two-way ANOVA (unpaired Student’s t tests for two population comparisons) or one-way ANOVA (Bonferroni’s multiple comparison test) using GraphPad Prism software (GraphPad Prism Software). The results are presented as means ± SEM and *p*-values less than 0.05 (*p* < 0.05) were considered statistically significant. Details on behavioral assays are described in Figure legends and Appendix A. Principal component analysis (PCA) was conducted on brain weight, behavior analysis data, neurogenesis analysis parameters, and estrogenic capacity that was most effective in discriminating between vehicle and D4-treated mice by using R software (version 4.0.3). The factoextra and biplot packages of R were used.

## Figures and Tables

**Figure 1 ijms-22-12949-f001:**
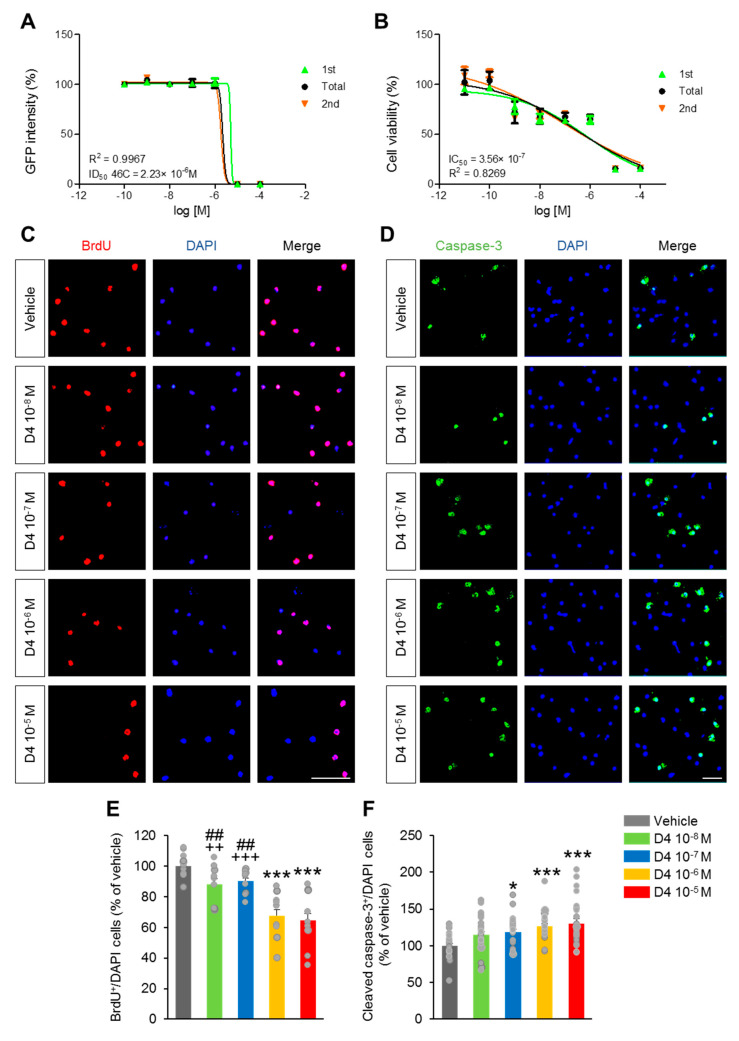
Cytotoxic effect of D4 on Sox1-GFP mouse embryonic stem cells and primary cortical neuron cells. (**A**) CCK8 assays were used to detect the difference in the viability of Sox1-GFP mouse embryonic stem cells after 48 h of treatment with different doses of D4. (**B**) The resultant intensity of integral GFP was observed after 96 h of treatment with different doses of D4. The IC50 and ID50 effects of D4 increased in strength with increasing dose. (**C**) Primary cortical neuronal cells were treated with D4 at DIV 1 and incubated with 10 μM BrdU for 2 h, 12 h after adding D4. Cells were fixed and immunofluorescence with BrdU antibodies (red). Scale bar: 40 µm. (**E**) Quantification of (**C**). The percentage of BrdU^+^ cells were decreased in the D4 10^−6^ M and D4 10^−5^ M groups compared to vehicle, D4 10^−8^ M and D4 10^−7^ M groups (F_4,100_ = 37.50, *p* < 0.0001). (**D**) Cell death was assessed by staining with a cleaved caspase-3 antibody (green) 12 h after administration of D4. Scale bar: 100 µm. (**F**) Quantification of (**D**). D4-treated groups, except D4 10^−8^ M group, showed higher in the percentage of cleaved caspase-3^+^ cells compared to vehicle group, (F_4,128_ = 7.614, *p* < 0.0001). *n* = 5 cell culture replicates using 5 mice for each condition (cell counts: 1000 cells for each group). (**G**) Representative images of cultured mouse primary cortical neuron at DIV 4 stained with antibodies against MAP2 (red) and Tau1 (green). Scale bar: 100 µm. (**H**–**M**) Quantification of relative number and length of dendrite and axon. (**H**) The number of primary dendrites, (**I**) number of secondary dendrites, (**J**) average dendrite length, (**K**) number of primary axons, (**L**) number of secondary axons, (**M**) average axon length. D4-treated groups showed no significant difference in the number and the length of dendrite and axon. Data represent mean ± SEM. Statistical significance was determined by one-way ANOVA with Bonferroni correction. * *p* < 0.05 vs. vehicle, *** *p* < 0.001 vs. vehicle, ^##^
*p* < 0.01 vs. 10^−7^ mol/L, ^++^
*p* < 0.01 vs. 10^−5^ mol/L, ^+++^
*p* < 0.001 vs. 10^−5^ mol/L. Treatments: vehicle; 0.1% DMSO, D4; 10^−8^ mol/L, 10^−7^ mol/L, 10^−6^ mol/L, or 10^−5^ mol/L.

**Figure 2 ijms-22-12949-f002:**
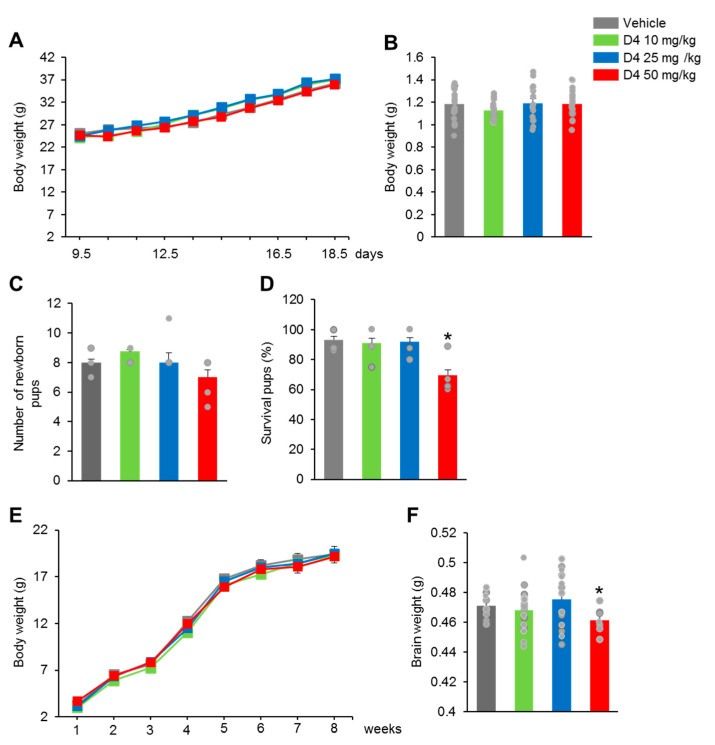
Prenatal D4 treatment reduced the survival of pups. (**A**) D4 treatment showed no changed in the body weight of the dams (*n* = 5–6 per group) (E19.5, F_3,18_ = 0.9215, *p* = 0.4543). (**B**) The body weight on PND1 (*n* = 20 per group) (F_3,74_ = 1.049, *p* = 0.3763) and (**C**) the number of newborn pups (*n* = 5–6 per group) (F_3,19_ = 0.7416, *p* = 0.5427) were shown. (**D**) The survival of pups was significantly decreased in the D4 50 mg/kg group (*n* = 40 per group) (F_3,19_ = 4.327, *p* = 0.0276). (**E**) There was no change in the body weight of the pups after birth (*n* = 40 per group) (PND28, F_3,94_ = 2.534, *p* = 0.0205; PND56, F_3,94_ = 1.223, *p* = 0.3072). (**F**) However, the brain weight was decreased in the D4 50 mg/kg group compared to vehicle group (*n* = 10 per group) (F_3,38_ = 3.568, *p* = 0.0237). Data represent mean ± SEM. Statistical significance was determined by one-way ANOVA with Bonferroni correction. * *p* < 0.05 vs. vehicle. Treatments: corn oil; vehicle, D4; 10 mg/kg/day, D4; 25 mg/kg, D4; 50 mg/kg.

**Figure 3 ijms-22-12949-f003:**
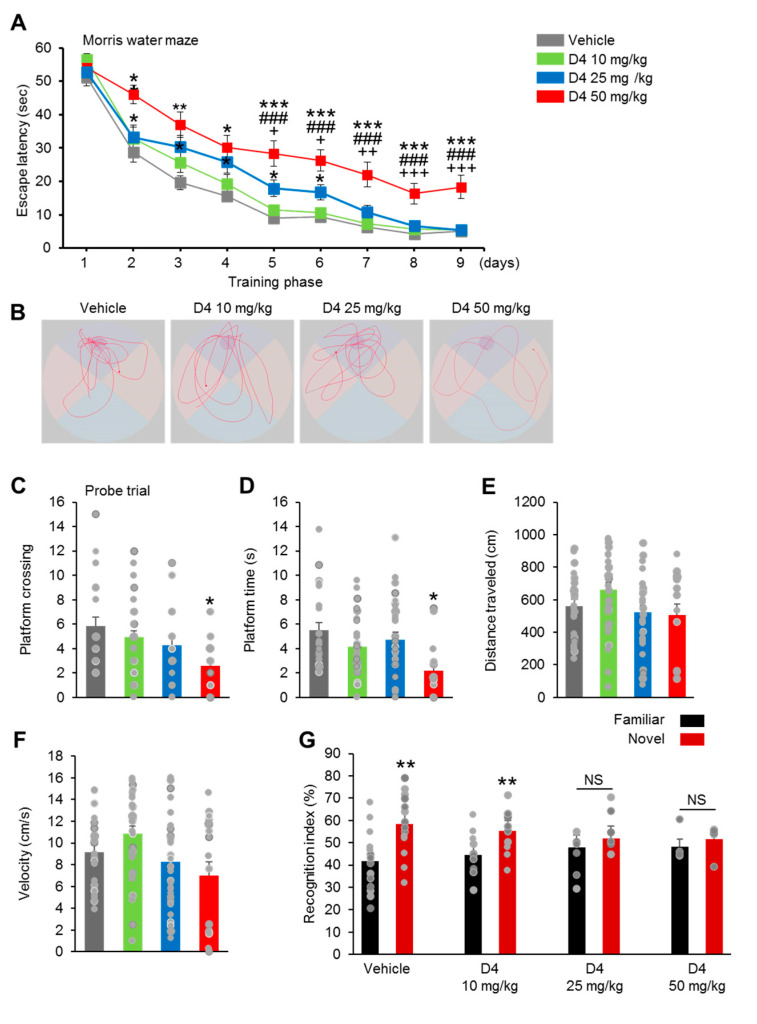
Prenatal D4 treatment induced behavioral deficits in adulthood. (**A**) In Morris water maze test, D4-treated groups exhibit spatial learning disability. (**B**) Representative swimming paths of vehicle and D4-treated mice during a probe trial. (**C**–**F**) Quantification of (**B**). D4 50 mg/kg group showed a slightly lower success in platform crossings and spent less time on platform. D4-treated groups showed no change in the swimming distances and swimming speeds compared to vehicle group. (**G**) In novel test, vehicle and D4 10 mg/kg groups displayed more time approaching and in proximity to the novel object. However, D4 25 mg/kg and D4 50 mg/kg groups displayed recognitive impairment. (**H**) D4-treated mice showed a decreased latency to fall during the rotarod test. (**I**) D4-treated mice exhibited normal in grip strength test. (**J**) D4-treated mice exhibited normal in social behavior in the three-chamber test. (**K**) D4-treated group preference index was no different compared to vehicle group. (**L**) Moreover, unlike vehicle and D4 mg/kg groups, D4 25 mg/kg and D4 50 mg/kg groups displayed impaired social novelty recognition by demonstrating no preference for a novel mouse over a familiar mouse. (**M**) D4 50 mg/kg group preference index was markedly lower compared to vehicle group. (**N**) In the forced swimming test, D4 25 mg/kg and D4 50 mg/kg groups showed higher in the immobility time compared to the vehicle group. (**O**) In the marble-burying test, compared with vehicle group, D4-treated groups showed a significantly greater number of marble burying bouts. Data represent mean ± SEM. Statistical significance was determined by one-way ANOVA with Bonferroni correction for multiple comparison test or two-tailed Student’s *t* test to compare two groups. * *p* < 0.05 vs. vehicle, ** *p* < 0.01 vs. vehicle, *** *p* < 0.001 vs. vehicle, ^#^
*p* < 0.05 D4 10 mg/kg vs. D4 50 mg/kg, ^##^
*p* < 0.01 D4 10 mg/kg vs. D4 50 mg/kg, ^###^
*p* < 0.001 D4 10 mg/kg vs. D4 50 mg/kg, ^+^
*p* < 0.05 D4 25 mg/kg vs. D4 50 mg/kg. ^++^
*p* < 0.01 D4 25 mg/kg vs. D4 50 mg/kg, ^+++^
*p* < 0.001 D4 25 mg/kg vs. D4 50 mg/kg.

**Figure 4 ijms-22-12949-f004:**
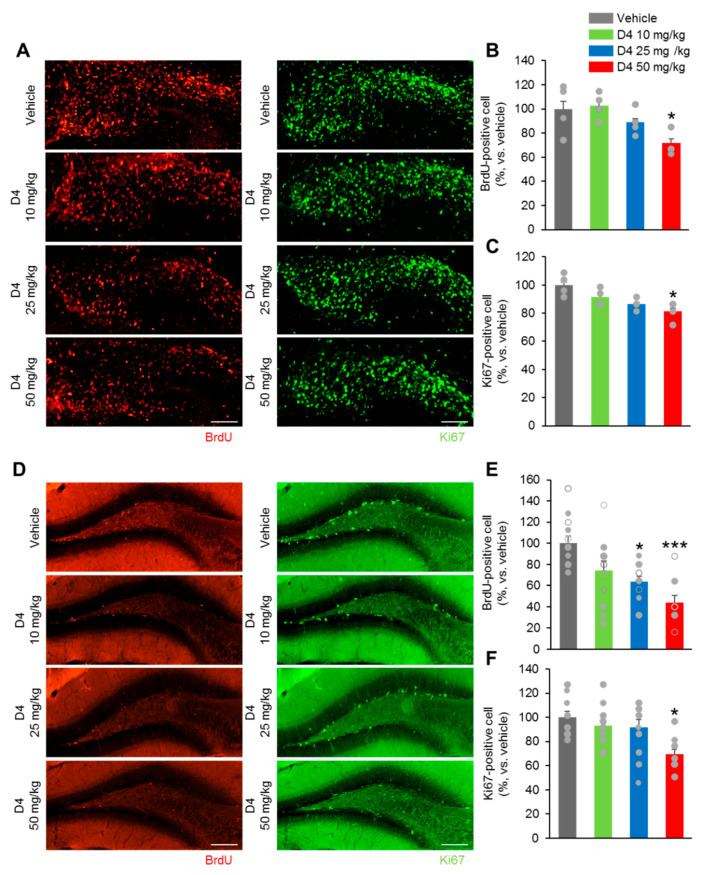
Prenatal D4 impairs neural progenitor cell premature differentiation and maintenance in vivo. Nuclei of cells in S phase of cell cycle in confocal sections of the dentate gyrus area are labeled using BrdU antibody (red) and progenitor cells are labeled using Ki67 (green) on confocal sections of the dentate gyrus area. Brains of vehicle and D4-treated groups at E17.5 (**A**) and offspring mice (**D**) were dissected. (**B**,**C**) Quantification analysis of (**A**). (**B**) The number of BrdU^+^ cells were decreased in the D4 25 mg/kg and D4 50 mg/kg groups compared to vehicle group at E18.5 (F_3,25_ = 5.840, *p* = 0.0143). (**C**) The number of Ki67^+^ cells were decreased in the D4 25 mg/kg and D4 50 mg/kg groups compared to vehicle group (F_3,25_ = 6.215, *p* = 0.0086), (*n* = 5–7 per group). (**E**,**F**) Quantification analysis of (**D**). (**E**) The number of BrdU^+^ cells were also decreased in the D4 25 mg/kg and D4 50 mg/kg groups compared to vehicle group (F_3,27_ = 6.689, *p* = 0.0002). (**F**) D4 25 mg/kg and D4 50 mg/kg groups displayed lower in the number of Ki67^+^ cells (F_3,24_ = 5.709, *p* = 0.0007), (*n* = 7–10 per group). The cell cycle exit rate was determined by dividing the number of BrdU-positive/Ki67-negative cells by the total number of BrdU-positive cells and the cell cycle re-entering was determined by dividing the number of BrdU-positive/Ki67-positive cells by the total number of BrdU-positive cells. At E17.5 dentate gyrus (**G**–**I**), the proportion of cells exiting the cell cycle in the D4 25 mg/kg and D4 50 mg/kg groups was markedly higher in the vehicle group (F_3,25_ = 6.487, *p* = 0.0036). D4 25 mg/kg and D4 50 mg/kg groups showed lower in the number of cells re-entering the cell cycle compared to the vehicle group (F_3,25_ = 4.627, *p* = 0.0251). At adult dentate gyrus area (**J**–**L**), D4 25 mg/kg and D4 50 mg/kg groups showed increased proportion of cells exiting the cell cycle compared to vehicle group (F_3,27_ = 8.308, *p* < 0.0001). Moreover, the number of cells re-entering the cell cycle in the D4 25 mg/kg and D4 50 mg/kg groups were lower compared to vehicle group (F_3,27_ = 8.500, *p* < 0.0001). *n* = 7–10 per group. Data represent mean ± SEM. Statistical significance was determined by one-way ANOVA with Bonferroni correction. * *p* < 0.05 vs. vehicle, *** *p* < 0.001 vs. vehicle. Scale bars: 50 µm.

**Figure 5 ijms-22-12949-f005:**
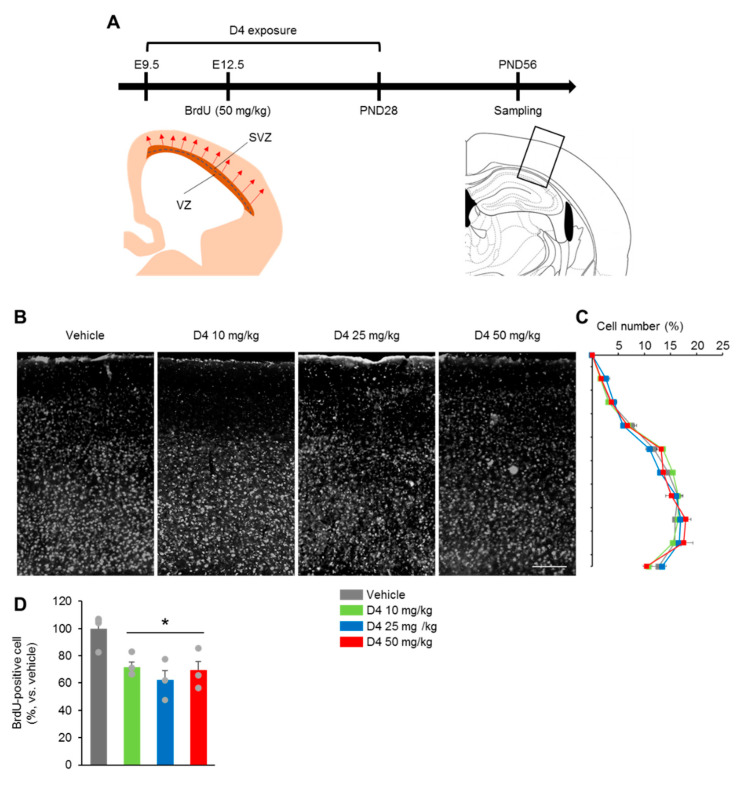
Prenatal D4 impaired neurogenesis but not the neural migration. (**A**) The pregnant mice received a single BrdU injection on E12.5, 3 h after the D4 administration to label mitotic progenitors in the embryonic brains. The brains of adult offspring mice were collected to examine cell migration. (**B**) The percentage of BrdU^+^ cells at each distance from the apical side is shown. There was no difference in the distribution of BrdU^+^ cells in the cerebral cortex (F_3,22_ = 1.406, *p* = 0.2718). (**C**) The numbers of BrdU^+^ cells in the tangential direction were shown. (**D**) The numbers of BrdU^+^ were lower in the D4-treated groups than in vehicle group (F_3,22_ = 6.855, *p* = 0.086). *n* = 5–6 per group. Data represent mean ± SEM. Statistical significance was determined by one-way ANOVA with Bonferroni correction. * *p* < 0.05 vs. vehicle. Scale bars: 200 µm in (**B**).

**Figure 6 ijms-22-12949-f006:**
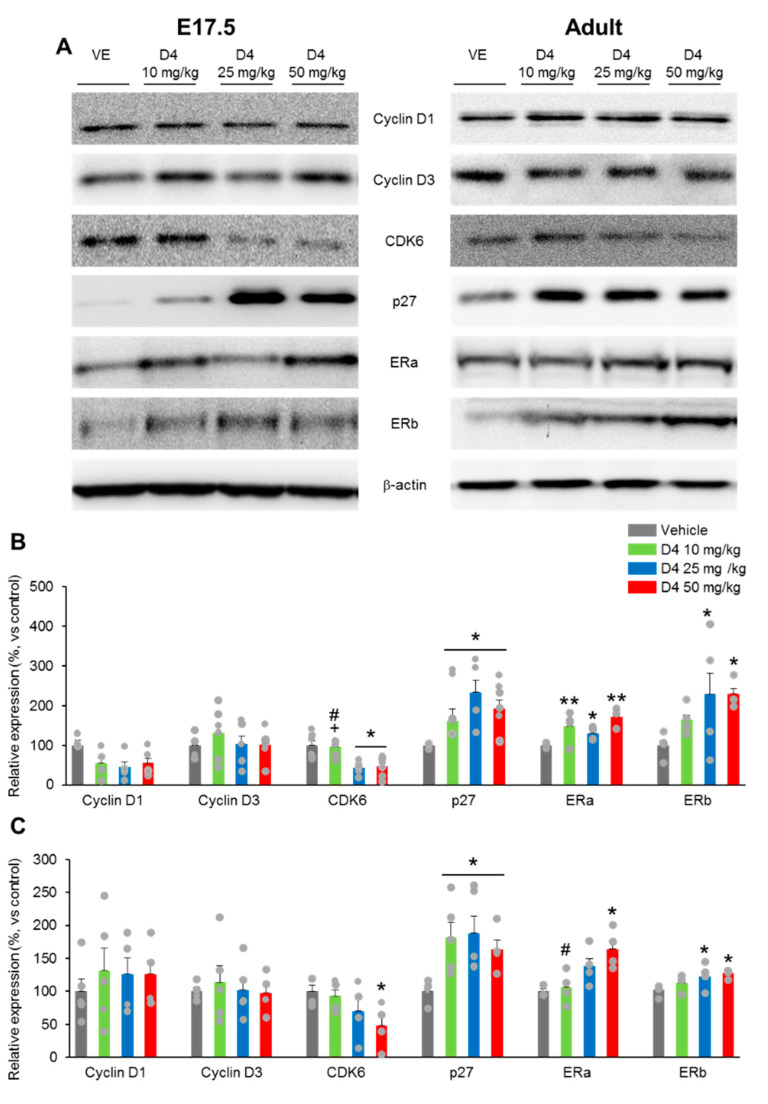
Prenatal D4 regulated cell cycle regulatory protein and estrogen receptor expression in neural progenitor cells. (**A**) Cell cycle regulatory protein content were analyzed by using Western blotting (β-actin as loading control). (**B**,**C**) Quantification of (**A**). (**B**) At E17.5, D4 25 mg/kg and 50 mg/kg groups showed significant decreases in CDK6 compared to both vehicle and D4 50 mg/kg groups (F_3,22_ = 8.886, *p* = 0.0007). Moreover, compared to vehicle group, D4-treated groups exhibited increases in p27 (F_3,22_ = 3.374, *p* = 0.0465). *n* = 5–7 per group. (**C**) At adult stage, D4 50 mg/kg groups showed decreased in CDK6 (F_3,20_ = 3.318, *p* = 0.0418). p27 was markedly higher in the D4-treated groups (F_3,20_ = 11.39, *p* = 0.0002). D4-treated groups showed increases in ERa at both E17.5 (F_3,27_ = 9.612, *p* = 0.0002) and adult stage (F_3,22_ = 5.194, *p* = 0.0177). However, compared to vehicle groups, D4 25 mg/kg and 50 mg/kg group exhibited increases in ERb E17.5; F_3,29_ = 4.406, *p* = 0.0124; adult; F_3,27_ = 5.218, *p* = 0.0065). *n* = 5 per group. Data represent mean ± SEM. Statistical significance was determined by one-way ANOVA with Bonferroni correction. * *p* < 0.05 vs. vehicle, ** *p* < 0.01 vs. vehicle, ^#^
*p* < 0.05 D4 10 mg/kg vs. D4 50 mg/kg, ^+^
*p* < 0.05 D4 25 mg/kg vs. D4 50 mg/kg.

**Figure 7 ijms-22-12949-f007:**
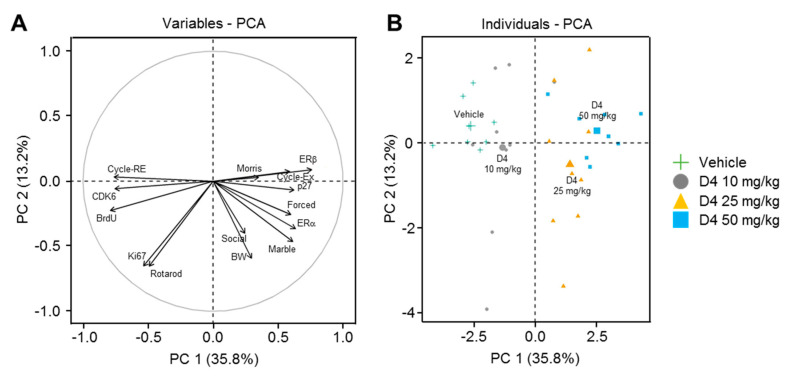
Multivariate principal component analysis (PCA) showing the effects of prenatal exposure to D4 on brain development. (**A**) The correlation circle. (**B**) The scatter plot. Cycle RE = cell cycle re-entry, Cycle Ex = cell cycle exit, ER alpha = ERa, ER beta = ERb, Morris = Morris water maze test, Forced = forced swimming test, Marble = marble burying test, BW = brain weight, Social = social novelty test, Rotarod = rotarod test.

## Data Availability

The datasets generated during and/or analyzed during the current study are available from the corresponding author on reasonable request.

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
