# Peer review of "Prenatal Octamethylcyclotetrasiloxane Exposure Impaired Proliferation of Neuronal Progenitor, Leading to Motor, Cognition, Social and Behavioral Functions"

_ijms, 2021, doi:10.3390/ijms222312949_

Round 1

Reviewer 1 Report

The authors of the paper entitled: "Prenatal octamethylcyclotetrasiloxane exposure impaired proliferation of neuronal progenitor, leading to motor, cognition, social and behavioral functions" aimed to investigate the potential neurotoxicity of D4 using in vitro and in vivo models. The novelty of the paper is good, the materials and methods are appropriate and the experimental plan is conducted adequately, the results are clearly presented and well discussed.
For these reasons, I suggest the acceptance of the manuscript in the present form. 

Author Response

The main question addressed by the research is related to the potential neurotoxicity of the emerging contaminant octamethylcyclotetrasiloxane through an in vivo approach. The relevance of the research topic is high, the novelty is good since there are few data aimed to investigate the developmental toxicity of octamethylcyclotetrasiloxane and also the potential mechanisms of action involved in such type of toxicity. The paper is well written, the introduction is pertinent, the materials and methods are robust and well described, as well as the results and the discussion, all aimed to reach valuable conclusions. 

We would like to thank you for your comment.

Reviewer 2 Report

This study describes the effects of pregnant mice and their offspring mice given daily with D4 from day 9.5 to day 28 of life. The authors collaborate on aberrant gene expression in prenatal D4 treated mice, such as the cyclin-dependent kinase CDK6, where D4 regulates the cell cycle, and the cyclin-dependent kinase inhibitor p27, and estrogen receptors. ERa and ERb were increased in the brains of prenatal D4 treated mice. These findings suggest that D4 exerts estrogenic activity that affects cell cycle progression of neural progenitor cells during neurogenesis, which may be associated with cognitive impairment in offspring.

The current article contains a novel and scientific sound of the new EDC. However, some parts need to be modified and rewritten before it can be published.

  1. The first sentence in Abstract, “Cyclic siloxane octamethylcyclotetrasiloxane (D4) is an endocrine-disrupting chemical (EDC)”

Authors have elucidated that D4 is an endocrine-disrupting chemical (EDC) in this study.  Therefore, this sentence should be deleted or rewrite to “Cyclic siloxane octamethylcyclotetrasiloxane (D4)has been concerned to be EDC”.

  1. Figure 7 and Conclusions should be rewritten because authors wrote that “Prenatal D4 regulated cell cycle regulatory protein in neural progenitor cells via estrogenic acitivity.”

However, authors determined protein expression Era and Erb only.  Author did not examine cell cycle-related gene under the conditions of siRNA assay or specific anti Era and Erb drug treatment.  Therefore, “via estrogenic acitivity” is over stated.

Author Response

We would like to thank the reviewers for their constructive suggestions on how to improve the quality of this manuscript. In the revised version, we have addressed the concerns that the reviewers raised.

This study describes the effects of pregnant mice and their offspring mice given daily with D4 from day 9.5 to day 28 of life. The authors collaborate on aberrant gene expression in prenatal D4 treated mice, such as the cyclin-dependent kinase CDK6, where D4 regulates the cell cycle, and the cyclin-dependent kinase inhibitor p27, and estrogen receptors. ERa and ERb were increased in the brains of prenatal D4 treated mice. These findings suggest that D4 exerts estrogenic activity that affects cell cycle progression of neural progenitor cells during neurogenesis, which may be associated with cognitive impairment in offspring.

The current article contains a novel and scientific sound of the new EDC. However, some parts need to be modified and rewritten before it can be published.

  1. The first sentence in Abstract, “Cyclic siloxane octamethylcyclotetrasiloxane (D4) is an endocrine-disrupting chemical (EDC)”

Authors have elucidated that D4 is an endocrine-disrupting chemical (EDC) in this study.  Therefore, this sentence should be deleted or rewrite to “Cyclic siloxane octamethylcyclotetrasiloxane (D4) has been concerned to be EDC”.

Response: First, we would like to thank you for your comment. As your suggestion, we changed this sentence in line 15-17:” Cyclic siloxane octamethylcyclotetrasiloxane (D4) has been concerned to be endocrine-disrupting chemical (EDC). D4 is widely used in detergent products, cosmetics, and personal care products.”

2. Figure 7 and Conclusions should be rewritten because authors wrote that “Prenatal D4 regulated cell cycle regulatory protein in neural progenitor cells via estrogenic acitivity.”

However, authors determined protein expression Era and Erb only.  Author did not examine cell cycle-related gene under the conditions of siRNA assay or specific anti Era and Erb drug treatment. Therefore, “via estrogenic acitivity” is over stated.

Response: As the reviewer indicated, we changed these sentences in:

- Line 396: “Figure 6. Prenatal D4 regulated cell cycle regulatory protein and estrogen receptor expression in neural progenitor cells.”

- Line 556-558: “In conclusion, D4 increases the expression of estrogen receptors, inhibits the cell cycle progression of neuronal progenitor cells by regulating the expression of cyclin-dependent kinases CDK6 and cyclin-dependent kinase inhibitor p27.”

We also added more information in line 559: “Furthermore, D4 has weak estrogenic activity, and that these effects are mediated through estrogen receptor [23, 46].”

Reviewer 3 Report

To the authors,
The effect of D4 was presented in this paper through an animal model.
The authors performed many experiments, organized the data well, and came up with interesting results. This is a result suitable for IJMS journal (IF=5.923).
However, it will be possible to publish only after some minor points have been corrected.

1. In Figure 1A, 1B, present each group (1st, Total, 2nd) in different colors.

2. Correct the abnormal spacing. For example, line 103, 106, 431. Paragraph classification is also need (Supplemental Methods; “Novel object recognition test”, “Nesting test”). Request a professional English proofreading company for full proofreading.

3. Please express the outlier mark as full-circle. However, it should not be confused with the statistical significance indication.

4. The description in the figure legend is too long (especially the legend in figure 3; lines 201-151). Organize the mean and SD for each group and present it as a supplementary table.

5. Figures 4 and 5 contain similar contents, so it is better to merge them. And the abbreviations in figure legends should be as self-fulling as possible. Please provide the full name of the abbreviation for each figure legend (e.g. DG area in Figure 4 legend).

6. It is sufficient to provide the manufacturer information for the program used when it was first released (e.g. lines 581 and 656).
7. Please provide the name of the R package used for analysis and, if possible, the name of the function used.

Author Response

We would like to thank the reviewers for their constructive suggestions on how to improve the quality of this manuscript. In the revised version, we have addressed the concerns that the reviewers raised.

The effect of D4 was presented in this paper through an animal model.

The authors performed many experiments, organized the data well, and came up with interesting results. This is a result suitable for IJMS journal (IF=5.923).

However, it will be possible to publish only after some minor points have been corrected.

1. In Figure 1A, 1B, present each group (1st, Total, 2nd) in different colors.

Response: First, we would like to thank you for your comment. As your suggestion, we present each group (1st, Total, 2nd) in different colors. We replaced Figure 1A and 1B in line 98.

2. Correct the abnormal spacing. For example, line 103, 106, 431. Paragraph classification is also need (Supplemental Methods; “Novel object recognition test”, “Nesting test”). Request a professional English proofreading company for full proofreading.

Response: We apologize for these errors. We have fixed these errors as the reviewer indications. Again, we have checked this manuscript and Supplemental Methods. This manuscript was proofread and edited by one of the English editors at company NURISCO.

3. Please express the outlier mark as full-circle. However, it should not be confused with the statistical significance indication.

Response: We have changed the outlier mark as full-circle in all Figures and Supplementary Figures as the reviewer’s suggestion.

4. The description in the figure legend is too long (especially the legend in figure 3; lines 201-151). Organize the mean and SD for each group and present it as a supplementary table.

Response: As the reviewer indications, we organized the mean and SD for each group and present it at Supplementary table 1. We have also improved the legend in figure 3; lines 206-259: “Figure 3. Prenatal D4-treatment induced behavioral deficits in adulthood. (A) In Morris water maze test, D4-treated groups exhibit spatial learning disability (B) Representative swimming paths of vehicle and D4-treated mice during a probe trial. (C, D, E, and F) Quantification of (B). D4 50 mg/kg group showed a slightly lower in platform crossings and spent less time in platform. D4-treated groups showed no change in the swimming distances and swimming speeds compared to vehicle group. (G) In novel test, vehicle and D4 10 mg/kg groups displayed more time approaching and in proximity to the novel object. However, D4 25 mg/kg and D4 50 mg/kg groups displayed shows recognitive impairment. (H) D4 -treated mice showed a decreased latency to fall during the rotarod test. (I) D4 -treated mice exhibited normal in grip strength test. (J) D4 mice exhibited normal in social behavior in the three-chamber test. (K) D4-treated groups preference index was no different compared to vehicle group. (L) Moreover, unlike vehicle and D4 mg/kg groups, D4 25 mg/kg and D4 50 mg/kg groups displayed impaired social novelty recognition by demonstrating no preference for a novel mouse over a familiar mouse. (M) D4 50 mg/kg group preference index was markedly lower compared to vehicle group. (N), In the forced swimming test, D4 25 mg/kg and D4 50 mg/kg groups showed higher in the immobility time compared to the vehicle group. (O) In the marble-burying test, compared with vehicle group, D4-treated groups showed a significantly greater number of marble burying bouts. Data represent mean ± SEM. Statistical significance was determined by one-way ANOVA with Bonferroni correction for multiple comparison test or two-tailed Student’s t test for to compare two groups. *p < 0.05 vs. vehicle, **p < 0.01 vs. vehicle, ***p < 0.001 vs. vehicle, #p < 0.05 D4 10 mg/kg vs. D4 50 mg/kg, ##p < 0.01 D4 10 mg/kg vs. D4 50 mg/kg, ###p < 0.001 D4 10 mg/kg vs. D4 50 mg/kg, +p < 0.05 D4 25 mg/kg vs. D4 50 mg/kg. ++p < 0.01 D4 25 mg/kg vs. D4 50 mg/kg, +++p < 0.001 D4 25 mg/kg vs. D4 50 mg/kg.”

5. Figures 4 and 5 contain similar contents, so it is better to merge them. And the abbreviations in figure legends should be as self-fulling as possible. Please provide the full name of the abbreviation for each figure legend (e.g. DG area in Figure 4 legend).

Response: Thank you for your suggestion. We merged Figures 4 and 5. We changed the legend of Figure 4 as follow: “Figure 4. Prenatal D4 impairs neural progenitor cell premature differentiation and maintenance in vivo. Nuclei of cells in S phase of cell cycle in confocal sections of the dentate gyrus area are labeled using BrdU antibody (red) and progenitor cells are labeled using Ki67 (green) on confocal sections of the dentate gyrus area. Brains of vehicle and D4-treated groups at E17.5 (A) and offspring mice (D) were dissected. (B, C) Quantification analysis of (A). (B) The number of BrdU+ cells were decreased in the D4 25 mg/kg and D4 50 mg/kg groups compared to vehicle group at E18.5 (F3,25 = 5.840, p = 0.0143). (C) The number of Ki67+ cells were decreased in the D4 25 mg/kg and D4 50 mg/kg groups compared to vehicle group (F3,25 = 6.215, p = 0.0086). (n = 5-7 per group). (E, F) Quantification analysis of (D). (E) The number of BrdU+ cells were also decreased in the D4 25 mg/kg and D4 50 mg/kg groups compared to vehicle group (F3,27 = 6.689, p = 0.0002). (F) D4 25 mg/kg and D4 50 mg/kg groups displayed lower in the number of Ki67+ cells (F3,24 = 5.709, p = 0.0007). (n = 7-10 per group). The cell cycle exit rate was determined by dividing the number of BrdU-positive/Ki67-negative cells by the total number of BrdU-positive cells and the cell cycle re-entering was determined by dividing the number of BrdU-positive/Ki67-positive cells by the total number of BrdU-positive cells. At E17.5 dentate gyrus (G, H, I), the proportion of cells exiting the cell cycle in the D4 25 mg/kg and D4 50 mg/kg groups was markedly higher in the vehicle group (F3,25 = 6.487, p = 0.0036). D4 25 mg/kg and D4 50 mg/kg groups showed lower in the number of cells re-entering the cell cycle compared to the vehicle group (F3,25 = 4.627, p = 0.0251). At adult dentate gyrus area (J, K, L) D4 25 mg/kg and D4 50 mg/kg groups showed increased proportion of cells exiting the cell cycle compared to vehicle group (F3,27 = 8.308, p < 0.0001). Moreover, the number of cells re-entering the cell cycle in the D4 25 mg/kg and D4 50 mg/kg groups were lower compared to vehicle group (F3,27 = 8.500, p < 0.0001). n = 7-10 per group. Data represent mean ± SEM. Statistical significance was determined by One-way ANOVA with Bonferroni correction. *p < 0.05 vs. vehicle, ***p < 0.001 vs. vehicle. Scale bars: 50 µm.”

Line 297-315, we also at the full name of the abbreviation for each figure legend in line 292.

6. It is sufficient to provide the manufacturer information for the program used when it was first released (e.g. lines 581 and 656).

Response: We added more information as the reviewer suggestion from line 562 to 687.

7. Please provide the name of the R package used for analysis and, if possible, the name of the function used.

Response: We added the name of the R package used for analysis in the Material and Methods, line 678-679: “The factoextra and biplot packages of R were used”.